# RETHINKING COUNTERFACTUAL FAIRNESS: ON WHICH INDIVIDUALS TO ENFORCE, AND HOW?

## ABSTRACT

Fairness in human and algorithmic decision-making is crucial in areas such as criminal justice, education, and social welfare. Recently, counterfactual fairness has drawn increasing research interest, suggesting that decision-making for individuals should remain the same when intervening with different values on the protected attributes. Nevertheless, the question of *"which attributes and individuals should be protected"* is rarely discussed in the existing counterfactual fairness literature. For example, when considering leg disability as a protected attribute, the algorithms should not treat individuals with leg disabilities differently in college admissions, but one may naturally take into this factor for the purpose of selecting runner athletes. In other words, when and how to enforce fairness is expected to depend on the causal relation between the protected attribute and the outcome of interest. Formally, this paper proposes *principal counterfactual fairness* using the concept of principal stratification from the causal inference literature, focusing on whether an algorithm is counterfactually fair for individuals whose protected attribute has no individual causal effect on the outcome of interest. To examine whether an algorithm satisfies principal counterfactual fairness, we derive the statistical bounds, and propose a post-processing approach to achieving principal counterfactual fairness with minimal individual decision changes. Experiments are conducted using synthetic and real-world datasets to verify the effectiveness of our methods.

## 1 INTRODUCTION

Addressing the fairness of automated algorithms is critical to making safe decisions in areas such as criminal justice (Brennan et al., 2009; Dieterich et al., 2016), education (Reardon and Owens, 2014), and social welfare (Chouldechova et al., 2018). To achieve fair machine learning, many association-based fairness notions have been proposed to constrain the statistical independence between protected attributes and decisions, e.g., statistical parity (Dwork et al., 2012), equalized odds (Hardt et al., 2016), and predictive parity (Chouldechova, 2017). In addition, the algorithmic fairness can also be approached from a causal perspective (Kusner et al., 2017; Zhang et al., 2017a;b; 2018a;b; Zhang and Bareinboim, 2018; Nabi and Shpitser, 2018; Wu et al., 2019a;b; Chiappa, 2019; Imai and Jiang, 2020; Mishler et al., 2021; Zuo et al., 2022). Among them, counterfactual fairness (Kusner et al., 2017) has garnered considerable attention recently. This criterion demands that any alterations made to the values of protected attributes do not result in changes to individual decision-making.

Nevertheless, as Chouldechova and Roth (2020) pointed out, the question of "which attributes and individuals should be protected" is rarely discussed in the existing counterfactual fairness literature. Should counterfactual fairness hold for *all* sensitive attributes on *all* individuals? For example, when considering leg disability as a protected attribute, it is reasonable to require that the algorithms should not treat individuals with disabilities differently in college admissions, but should the algorithms also be required to make the same decisions for individuals with disabilities when selecting runner athletes? In such cases, it is clear that it is not appropriate to select individuals with disabilities as running athletes. But what if the sensitive attribute is gender or race instead? How to reflect the differences between disability and gender as sensitive attributes?

To tackle the above issues, we summarize relevant studies in Table 1, which can be broadly divided into two branches. On one hand, instead of requiring fairness to hold on all individuals as in demographic parity (Darlington, 1971), equalized odds (Hardt et al., 2016) constrains the examination

Table 1: A summary of the proposed principal counterfactual fairness and related concepts.

| Fairness Definition | Formulation ($A$: protected attribute; $D$: decision; $Y$: outcome; $X$: covariate) |
| --- | --- |
| Demographic Parity (Darlington, 1971) | $A \perp\!\!\!\perp D$ |
| Equalized Odds (Hardt et al., 2016) | $A \perp\!\!\!\perp D \mid Y$ |
| Equality of Opportunity (Hardt et al., 2016) | $A \perp\!\!\!\perp D \mid Y = 1$ |
| Counterfactual Equalized Odds (Mishler et al., 2021) | $A \perp\!\!\!\perp D \mid Y(D=0) = 1$ |
| Principal Fairness (Imai and Jiang, 2020) | $A \perp\!\!\!\perp D \mid (Y(D=0), Y(D=1))$ |
| Counterfactual Parity (Mitchell et al., 2021) | $\mathbb{P}(D(0) = 1) = \mathbb{P}(D(1) = 1)$ |
| Conditional Counterfactual Fairness (Mitchell et al., 2021) | $\mathbb{P}(D(0) = 1) = \mathbb{P}(D(1) = 1) \mid X$ |
| Principal Counterfactual Parity (ours) | $\mathbb{P}(D(0) = 1) = \mathbb{P}(D(1) = 1) \mid Y(A=0) = Y(A=1)$ |
| Principal Conditional Counterfactual Fairness (ours) | $\mathbb{P}(D(0) = 1) = \mathbb{P}(D(1) = 1) \mid Y(A=0) = Y(A=1), X$ |
| Principal Counterfactual Equalized Odds (ours) | $\mathbb{P}(D(0) = 1) = \mathbb{P}(D(1) = 1) \mid Y(A=0) = Y(A=1) = y, X$ |
| Counterfactual Fairness (Kusner et al., 2017) | $D_i(A_i = 0) = D_i(A_i = 1)$ |
| Path-Specific Counterfactual Fairness (Chiappa, 2019) | $D_i(A_i = 0) = D_i(A_i = 1, M_i(A_i) = M_i(0))$ |
| Principal Counterfactual Fairness (ours) | $D_i(A_i = 0) = D_i(A_i = 1)$ holds for $Y_i(A_i = 0) = Y_i(A_i = 1)$ |

of demographic parity to subgroups with the same observed outcome. By considering the effect of decision-making on the observed outcomes, counterfactual equalized odds (Mishler et al., 2021) generalizes the above concepts to make fair decisions on individuals with the same value on the potential outcome under control. Principle fairness (Imai and Jiang, 2020) further uses the concept of principal stratification from the causal inference literature to consider the joint potential outcomes of the decision on outcome. However, despite considering specific subgroups defined from a counterfactual view, these fairness notions still use the *statistical independence* of sensitive attributes and decision-making on that *subgroup*, which is not sufficient to guarantee *individual* counterfactual fairness (Kusner et al., 2017; Mitchell et al., 2021). On the other hand, path-specific counterfactual fairness (Chiappa, 2019) promotes the notion of counterfactual fairness to restricted on unfair paths, rather than considering the total effect of sensitive attributes on decision-making. Despite partially answering the question of "which attributes should be protected", similar to counterfactual fairness, path-specific counterfactual fairness requires fairness on *all* individuals. This motivates us to rethink counterfactual fairness to better answer that "which attributes and individuals should be protected".

In this paper, instead of forcing decisions to remain the same for all individuals when the protected attribute changes as in counterfactual fairness, we propose *principal counterfactual fairness* using the concept of principal stratification from the causal inference literature (Frangakis and Rubin, 2002; Pearl, 2011), focusing on whether the counterfactual fairness holds for individuals whose protected attribute has no individual causal effect on the outcome of interest. For the aforementioned example, since leg disability (as a sensitive attribute) may affect athlete performance (as an outcome), we only require that decisions remain similar for those individuals with disabilities that do not affect athlete performance. In contrast, since disability and gender (as sensitive attributes) do not have a causal effect on the exam pass (as an outcome), we might expect decision-making for all individuals to satisfy counterfactual fairness. In summary, the proposed principal counterfactual fairness further considers the effect of protected attributes on outcomes of interest from a counterfactual perspective, and we show the principal counterfactual fairness would degenerate to standard counterfactual fairness when the protected attributes have no individual causal effect on outcomes for all individuals.

To examine whether an algorithm satisfies principal counterfactual fairness, we first derive the necessary conditions[1] for an algorithm to satisfy principal counterfactual fairness based on statistical bounds. Then we propose an optimization-based evaluation method to test whether an algorithm satisfies principal counterfactual fairness. Specifically, the algorithm does not satisfy the principal counterfactual fairness if the feasible region under particular constraints is the empty set, or if there exists a principal stratum with the optimized maximum probability value less than zero. We further propose a principled post-processing approach to achieve principal counterfactual fairness with minimal individual decision changes, and theoretically prove the optimality of the post-processing approach using doubly robust estimation. We conduct extensive experiments on synthetic and real-world datasets to verify the effectiveness of the proposed algorithm.

The main contributions of this paper are:

---

[1]Due to partial identifiability, it is difficult to find necessary and sufficient conditions for principal counterfactual fairness, similar problems also exist in the counterfactual fairness literature (Kusner et al., 2017).

- We propose a novel fairness notion using the concept of principal stratification, called *principal counterfactual fairness*, which requires the counterfactual fairness to hold only when the protected attribute has no individual causal effect on the outcome of interest.

- We derive the necessary conditions for an algorithm to satisfy principal counterfactual fairness based on statistical bounds, and propose an optimization-based evaluation method to test whether an algorithm satisfies principal counterfactual fairness.

- We further propose a principled post-processing approach to achieving principal counterfactual fairness with minimal individual decision changes, and theoretically prove the optimality of the post-processing approach using doubly robust estimation.

- We conduct experiments on both synthetic and real-world datasets to verify the effectiveness of the proposed optimization-based evaluation and post-processing approach.

## 2 PRELIMINARIES

We first formalize the issue of fairness in decision making, as well as summarize the related statistical and counterfactual fairness notions that have been widely studied. Suppose a simple random sample of $n$ units from a super population $\mathbb{P}$, for each unit $i$, the covariate (e.g., age or income) and the binary protected attribute (e.g., gender or disability) are denoted as $X_i \in \mathcal{X}$ and $A_i \in \{0, 1\}$, respectively. Let $Y_i \in \mathcal{Y} = \{0, 1\}$ be the binary outcome variable of interest and $D_i \in \{0, 1\}$ be the binary decision variable. For the simplicity of exposition, we assume the protected attribute, decision variable, and outcome variable are all binary, and covariates are discrete, but these variables can all be extended to other variable types in our work. To study the counterfactual fairness problem, we adopt the potential outcome framework (Rubin, 1974; Neyman, 1990). Specifically, let $Y_i(0)$ and $Y_i(1)$ be the outcome of the unit $i$ had this unit have the protected attribute $A_i = 0$ and $A_i = 1$, respectively. Since each unit can only have one particular value of protected attribute, we always observe the corresponding outcome be either $Y_i(0)$ or $Y_i(1)$, but not both. This is also known as the fundamental problem of causal inference (Holland, 1986; Morgan and Winship, 2015). Formally, the observed outcome for unit $i$ is $Y_i = (1 - A_i)Y_i(0) + A_iY_i(1)$. In other words, the observed outcome is the potential outcome corresponding to the protected attribute value, which is also known as the consistency assumption in the causal inference literature (Hernán and Robins, 2020).

Based on the observed protected attributes, covariates, and outcomes of interest, i.e., $\{(A_i, X_i, Y_i)\}_{i=1}^N$, a machine learning algorithm $D(\cdot)$ for decision-making is obtained. Specifically, let $D_i(0)$ and $D_i(1)$ be the potential algorithmic decisions for the unit $i$ had this unit have the protected attribute $A_i = 0$ and $A_i = 1$, respectively. By the consistency assumption again, the algorithmic decision for individual $i$ in the factual world would be $D_i$. In order for algorithms to make fair decisions, as shown in Table 1, many statistical fairness notions have been proposed, such as demographic parity (Darlington, 1971), i.e., $A \perp\!\!\!\perp D$, equalized odds (Hardt et al., 2016), i.e., $A \perp\!\!\!\perp D \mid Y$, and equality of opportunity (Hardt et al., 2016), i.e., $A \perp\!\!\!\perp D \mid Y = 1$. By noting the causal effect of decision $D$ on the observed outcomes $Y$, counterfactual equalized odds generalizes the above concepts to make fair decisions on individuals with counterfactual advantaged outcomes (Mishler et al., 2021), i.e., $A \perp\!\!\!\perp D \mid Y(D = 0) = 1$. Principle fairness further uses the concept of principal stratification from the causal inference literature to consider the joint potential outcome of decisions on outcomes (Imai and Jiang, 2020), i.e., $A \perp\!\!\!\perp D \mid (Y(D = 0), Y(D = 1))$. Nevertheless, despite considering a specific counterfactual stratum, these fairness notions still use the *statistical independence* of sensitive attributes and decisions on that *stratum*, which is not sufficient to guarantee *causal effect-based* fairness notions (Kusner et al., 2017; Mitchell et al., 2021).

Instead of considering the statistical (conditional) independence between the protected attribute $A$ and decision $D$, causality-based fairness considers the causal effect of the protected attribute $A$ on decision $D$. Among them, *counterfactual parity* in Definition 1 requires that there is no *average causal effect* of the protected attribute $A$ on decision $D$ over the population (Mitchell et al., 2021).

**Definition 1** (Counterfactual parity (Mitchell et al., 2021)). *An algorithm $D$ for decision-making satisfies counterfactual parity, if under any value $a$ and $a'$ attainable by $A$,*

$$\mathbb{P}(D(a) = 1) = \mathbb{P}(D(a') = 1).$$

Table 2: The principal counterfactual fairness considers units in the principal fairness strata (in red), whereas counterfactual fairness considers all units including in the auxiliary fairness strata (in blue).

| Observed data | $(A = 0, Y = 0)$ | $(A = 0, Y = 1)$ | $(A = 1, Y = 0)$ | $(A = 1, Y = 1)$ |
|---|---|---|---|---|
| Principal fairness | $(Y(0) = 0, Y(1) = 0)$ | $(Y(0) = 1, Y(1) = 1)$ | $(Y(0) = 0, Y(1) = 0)$ | $(Y(0) = 1, Y(1) = 1)$ |
| Auxiliary fairness | $(Y(0) = 0, Y(1) = 1)$ | $(Y(0) = 1, Y(1) = 0)$ | $(Y(0) = 1, Y(1) = 0)$ | $(Y(0) = 0, Y(1) = 1)$ |

By incorporating the covariate $X$, *conditional counterfactual fairness* in Definition 2 requires that there is no *conditional average causal effect* of the protected attribute $A$ on the decision $D$ over subpopulations under context $X = x$ for all $x \in \mathcal{X}$.

**Definition 2** (Conditional counterfactual fairness (Mitchell et al., 2021)). *An algorithm $D$ for decision-making is conditional counterfactually fair, if under any context $X = x$ and any value $a$ and $a'$ attainable by $A$,*

$$\mathbb{P}(D(a) = 1 \mid X = x) = \mathbb{P}(D(a') = 1 \mid X = x).$$

Different from *counterfactual parity* in Definition 1 which constrains on the *total population* and *conditional counterfactual fairness* in Definition 2 which constrains on the *subpopulations* determined by the covariates, individual *counterfactual fairness* in Definition 3 further requires that there is no *individual causal effect* of the protected attribute $A$ on the decision $D$ *over all the individuals.*

**Definition 3** (Counterfactual fairness (Kusner et al., 2017)). *An algorithm $D$ for decision-making is individual counterfactually fair[2], if under any context $X = x$ and any value $a$ and $a'$ attainable by $A$,*

$$\mathbb{P}(D_i(a) = D_i(a')) = 1.$$

Counterfactual fairness states that $A$ should not be a cause of decision $D$ in any individual instance, with many follow-up studies (Zhang and Bareinboim, 2018; Chiappa, 2019). As in Table 1, one representative variant is path-specific counterfactual fairness, which requires counterfactual fairness to hold only on unfair paths (Chiappa, 2019).

Despite partially answering the question of "which attributes should be protected", similar to counterfactual fairness, path-specific counterfactual fairness also requires fairness on *all* individuals. This motivates us to rethink these counterfactual fairness notions to better answer the question of "which and how to decide the attributes and individuals that should be protected".

## 3 PRINCIPAL COUNTERFACTUAL FAIRNESS

In this section, we first propose the notions of principal counterfactual fairness using the concept of principal stratification from the causal inference literature. Ordered from weakest to strongest, we propose principal counterfactual parity in Definition 4, principal conditional counterfactual fairness in Definition 5, principal counterfactual equalized odds in Definition 6, and principal conditional counterfactual fairness in Definition 7, respectively. We also derive the necessary conditions for an algorithmic decision to satisfy principal counterfactual fairness based on statistical bounds.

Specifically, the principal strata are defined as the joint potential outcome values (Frangakis and Rubin, 2002), i.e., $(Y_i(a), Y_i(a'))$, where $a$ and $a'$ are the sensitive attribute values attainable by $A$, and each principal stratum represents how an individual would be affected by the protected attribute on the outcome of interest. In the proposed principal counterfactual fairness, we focus on whether the counterfactual fairness notions hold on individuals whose protected attribute has no individual causal effect on the outcome of interest, i.e., $Y_i(a) = Y_i(a')$ for all $a$ and $a'$ attainable by $A$.

Compared with the previous counterfactual fairness notions, Table 2 shows the difference: the proposed principal counterfactual fairness notions focus only on those individuals in "principal fairness" stratum (in red), while previous counterfactual fairness notions focus on individuals in both "principal fairness" stratum (in red) and "auxiliary fairness" stratum (in blue). Unlike the observed outcome $Y_i$, however, the potential outcomes, and hence principal strata, are not affected by the

---

[2]Counterfactual fairness in Kusner et al. (2017) refers to individual counterfactual fairness with the definition that $\mathbb{P}(D_{A \leftarrow a}(U) = y \mid X = x, A = a) = \mathbb{P}(D_{A \leftarrow a'}(U) = y \mid X = x, A = a)$. This is equivalent to $\mathbb{P}(D_i(a) = D_i(a')) = 1$ using potential outcomes formulation (Mitchell et al., 2021).

sensitive attribute value. Moreover, since we only observe one potential outcome for any individual, principal strata are not directly observable and be distinguished, as shown in Table 2.

In the disabled athlete selection example, the principal strata are defined by the athlete performance $Y_i(A_i)$ under each of the two scenarios—disabled $A_i = 1$ or not disabled $A_i = 0$. Then it is fair to let the algorithmic decision $D$ be unaffected by whether those individuals are disabled $A = 1$ or not $A = 0$, because the disability of those individuals has no individual causal effect on the athlete's performance, i.e., $Y_i(0) = Y_i(1)$. The following Definition 4 formally states the principal counterfactual parity, which requires counterfactual parity to hold on that particular stratum.

**Definition 4** (Principal counterfactual parity). *An algorithm $D$ for decision-making satisfies principal counterfactual parity, if under any value $a$ and $a'$ attainable by $A$,*

$$\mathbb{P}(D(a) = 1 \mid Y(a) = Y(a')) = \mathbb{P}(D(a') = 1 \mid Y(a) = Y(a')).$$

By conditional on covariate $X$, Definition 5 states the principal conditional counterfactual fairness.

**Definition 5** (Principal conditional counterfactual fairness). *An algorithm $D$ for decision-making is principal conditional counterfactually fair, if under any context $X = x$ and any value $a$ and $a'$ attainable by $A$,*

$$\mathbb{P}(D(a) = 1 \mid Y(a) = Y(a'), X = x) = \mathbb{P}(D(a') = 1 \mid Y(a) = Y(a'), X = x).$$

We now describe the potential limitations of using principal conditional counterfactual fairness in Definition 5. Recall the disabled athlete selection example and let $Y$ denote athlete performance. Although the protected attribute has no individual causal effect on the outcome for both $Y(a) = Y(a') = 0$ and $Y(a) = Y(a') = 1$ individuals. However, since individuals at $Y(a) = Y(a') = 1$ have better athlete performance compared with individuals at $Y(a) = Y(a') = 0$, it is natural to allow high probability of being selected as an athlete for individuals at the stratum $Y(a) = Y(a') = 1$. That is, $\mathbb{P}(D(a) = 1 \mid Y(a) = Y(a') = 1) > \mathbb{P}(D(a) = 1 \mid Y(a) = Y(a') = 0)$. This motivates us to further divide stratum $(Y(0) = Y(1))$ into multiple strata $(Y(0) = Y(1) = y)$ for all $y \in \mathcal{Y}$, and propose the corresponding principal counterfactual equalized odds in Definition 6.

**Definition 6** (Principal counterfactual equalized odds). *An algorithm $D$ for decision-making satisfies principal counterfactual equalized odds, if under any context $X = x$ and any value $a$ and $a'$ attainable by $A$, for all $y \in \mathcal{Y}$,*

$$\mathbb{P}(D(a) = 1 \mid Y(a) = Y(a') = y, X = x) = \mathbb{P}(D(a') = 1 \mid Y(a) = Y(a') = y, X = x).$$

For the case of binary variables, it is equivalent to $\tau_0(x) = \tau_1(x) = 0$, where

$$\tau_y(x) = \mathbb{P}(D(1) = 1 \mid Y(0) = Y(1) = y, X = x) - \mathbb{P}(D(0) = 1 \mid Y(0) = Y(1) = y, X = x),$$

for $y = 0, 1$. Denote $p_{ay}(x) = \mathbb{P}(Y = y \mid A = a, X = x)$ and $q_{ay}(x) = \mathbb{P}(D = 1 \mid A = a, Y = y, X = x)$, which can be calculated from the observed data. Under the ignorability assumption, the following lemma provides the sharp bounds on $\tau_0(x)$ and $\tau_1(x)$.

**Assumption 1** (Ignorability). $A \perp\!\!\!\perp (Y(1), Y(0), D(1), D(0)) \mid X$.

**Lemma 1.** *Under Assumption 1, the sharp upper and lower bounds on $\tau_0(x)$ are*

$$\text{Lower}(\tau_0(x)) = \max\left\{0, 1 - \frac{(1 - q_{10}(x))p_{10}(x)}{p_{10}(x) - p_{01}(x)}\right\} - \min\left\{1, \frac{q_{00}(x)p_{00}(x)}{p_{10}(x) - p_{01}(x)}\right\},$$

$$\text{Upper}(\tau_0(x)) = \min\left\{1, \frac{q_{10}(x)p_{10}(x)}{p_{10}(x) - p_{01}(x)}\right\} + \min\left\{0, \frac{(1 - q_{00}(x))p_{00}(x)}{p_{10}(x) - p_{01}(x)} - 1\right\}.$$

*The sharp upper and lower bounds on $\tau_1(x)$ are*

$$\text{Lower}(\tau_1(x)) = \max\left\{0, 1 - \frac{(1 - q_{11}(x))p_{11}(x)}{p_{01}(x) - p_{10}(x)}\right\} - \min\left\{1, \frac{q_{01}(x)p_{01}(x)}{p_{01}(x) - p_{10}(x)}\right\},$$

$$\text{Upper}(\tau_1(x)) = \min\left\{1, \frac{q_{11}(x)p_{11}(x)}{p_{01}(x) - p_{10}(x)}\right\} + \min\left\{0, \frac{(1 - q_{01}(x))p_{01}(x)}{p_{01}(x) - p_{10}(x)} - 1\right\}.$$

The following Theorem 1 gives the necessary inequality conditions to determine whether the algorithm satisfies *principle counterfactual equalized odds* based on statistical bounds in Lemma 1.

**Theorem 1.** *Under Assumption 1, the principle counterfactual equalized odds in Definition 6 under stratum $Y(0) = Y(1) = 0$ is violated if either of the following two inequalities holds:*

$$q_{00}(x)p_{00}(x) + (1 - q_{10}(x))p_{10}(x) < p_{10}(x) - p_{01}(x),$$
$$q_{10}(x)p_{10}(x) + (1 - q_{00}(x))p_{00}(x) < p_{10}(x) - p_{01}(x).$$

*Similarly, the principle counterfactual equalized odds in Definition 6 under stratum $Y(0) = Y(1) = 1$ is violated if either of the following two inequalities holds:*

$$q_{01}(x)p_{01}(x) + (1 - q_{11}(x))p_{11}(x) < p_{01}(x) - p_{10}(x),$$
$$q_{11}(x)p_{11}(x) + (1 - q_{01}(x))p_{01}(x) < p_{01}(x) - p_{10}(x).$$

Further, instead focusing on a specific subgroup, as an extension of individual counterfactual fairness in (Kusner et al., 2017), we define principal counterfactual fairness to achieve strict individual fair.

**Definition 7** (Principal counterfactual fairness). *An algorithm $D$ for decision-making is principal counterfactually fair with respect to outcome $Y$, if under any value $a$ and $a'$ attainable by $A$,*

$$\mathbb{P}(D_i(a) = D_i(a') \mid Y_i(a) = Y_i(a')) = 1.$$

We finally point out that the principal counterfactual fairness would degenerate to counterfactual fairness when $Y_i(a) = Y_i(a')$ holds on all individuals for any value $a$ and $a'$ attainable by $A$.

**Corollary 1** (Relation to counterfactual fairness). *The principal counterfactual fairness is equivalent to counterfactual fairness in (Kusner et al., 2017), when the protected attributes have no individual causal effect on outcomes for all individuals.*

## 4 IMPLEMENTING PRINCIPAL COUNTERFACTUAL FAIRNESS

### 4.1 OPTIMIZATION-BASED EVALUATION

We started with an optimization-based evaluation method for principal counterfactual fairness in Definition 7, while other principal counterfactual fairness notions can be evaluated by similar arguments. Denote $w_{d_0,d_1,y_0,y_1}(x) = \mathbb{P}(D(0) = d_0, D(1) = d_1, Y(0) = y_0, Y(1) = y_1 \mid X = x)$, then principal counterfactual fairness is equivalent to $w_{0,1,0,0}(x) = w_{1,0,0,0}(x) = w_{0,1,1,1}(x) = w_{1,0,1,1}(x) = 0$. The proposed optimization constraints for evaluating whether the algorithmic decisions satisfy the principal counterfactual fairness are given as

$$w_{0,1,0,0}(x) = w_{1,0,0,0}(x) = w_{0,1,1,1}(x) = w_{1,0,1,1}(x) = 0,$$
$$w_{d_0,d_1,y_0,y_1}(x) \geq 0 \quad \text{for all} \quad d_0, d_1, y_0, y_1 \in \{0,1\},$$
$$\sum_{a,b} w_{d_0,a,y_0,b}(x) = \mathbb{P}(D(0) = d_0, Y(0) = y_0 \mid X = x) \quad \text{for all} \quad d_0, y_0 \in \{0,1\},$$
$$\sum_{a,b} w_{a,d_1,b,y_1}(x) = \mathbb{P}(D(1) = d_1, Y(1) = y_1 \mid X = x) \quad \text{for all} \quad d_1, y_1 \in \{0,1\},$$

for all $x \in \mathcal{X}$, where the first equation is the equivalent condition of principal counterfactual fairness, the second equation comes from the positivity of the probabilities, and the last two equations come from the definition of $w(x)$. Notably, under Assumption 1, the terms on the right-hand side of the last two equations can be identified and estimated using the observed data (see Section 4.2). Therefore, with these constraints on $w(x)$ imposed by the above equations set, we can determine that the algorithmic decision does not satisfy principal counterfactual fairness, if there exists $x \in \mathcal{X}$ such that the feasible domain of $w(x)$ satisfying these constraints is the empty set. In practice, we can also take one of $w_{0,1,0,0}(x)$, $w_{1,0,0,0}(x)$, $w_{0,1,1,1}(x)$, and $w_{1,0,1,1}(x)$, denoted as $\tilde{w}(x)$, as the objective function, and let the remaining three terms equal to zero as the optimization constraints, then solve the minimum and maximum values of $\tilde{w}(x)$, and obtain its value interval by solving this optimization problem. The algorithm should be considered as a violation of principal counterfactual fairness, when the minimum value of $\tilde{w}(x)$ is greater than 0 or the maximum value of $\tilde{w}(x)$ is less than 0.

### 4.2 ESTIMATION

Let $\mu_a^{d,y}(x) = \mathbb{P}(D = d, Y = y|A = a, X = x)$ and $\pi_a(x) = \mathbb{P}(A = a \mid X = x)$, with $\hat{\mu}_a^{d,y}(x)$ and $\hat{\pi}_a(x)$ be the estimated conditional-mean and propensity, respectively. To estimate the right-hand

side of the last two equations in the above optimization problem, without loss of generality, one needs to estimate $\mathbb{P}(D(a) = d, Y(a) = y \mid X = x)$ for $a, d, y \in \{0, 1\}$. Let $\hat{\mathbb{P}}$ and $\hat{\mathbb{E}}$ be the estimated probability and expectation that can be obtained via regression or subclassification. Then the outcome regression (OR) estimator is given as $\hat{\mathbb{P}}^{OR}(D(a) = d, Y(a) = y \mid X = x) = \hat{\mathbb{P}}(D = d, Y = y | A = a, X = x) = \hat{\mu}_a^{d,y}(x)$. The inverse propensity scoring (IPS) estimator is given as $\hat{\mathbb{P}}^{IPS}(D(a) = d, Y(a) = y \mid X = x) = \hat{\mathbb{E}}\left[\mathbb{I}(A = a) \cdot \mathbb{I}(D = d, Y = y)/\hat{\pi}_a(X)|X = x\right]$, and the doubly robust (DR) estimator is given as $\hat{\mathbb{P}}^{DR}(D(a) = d, Y(a) = y \mid X = x) = \hat{\mathbb{E}}\left[\hat{\mu}_a^{d,y}(X) + \mathbb{I}(A = a) \cdot \left(\mathbb{I}(D = d, Y = y) - \hat{\mu}_a^{d,y}(X)\right)/\hat{\pi}_a(X)|X = x\right]$.

**Theorem 2.** *Suppose that* $||\hat{\pi}_a(x) - \pi_a(x)||_2 \cdot ||\hat{\mu}_a^{d,y}(x) - \mu_a^{d,y}(x)||_2 = o_{\mathbb{P}}(n^{-1/2})$ *for all* $x \in \mathcal{X}$ *and $a$ attainable by $A$, then* $\hat{\mathbb{P}}^{DR}(D(a) = d, Y(a) = y \mid X = x)$ *is asymptotically normal*

$$\sqrt{n}\{\hat{\mathbb{P}}^{DR}(D(a) = d, Y(a) = y \mid X = x) - \mathbb{P}(D(a) = d, Y(a) = y \mid X = x)\} \to N(0, \sigma_1(x)^2),$$

*where* $\sigma_1(x)^2 = \mathbb{V}[\hat{\mu}_a^{d,y}(X) + \mathbb{I}(A = a) \cdot \left(\mathbb{I}(D = d, Y = y) - \hat{\mu}_a^{d,y}(X)\right)/\hat{\pi}_a(X)|X = x]$.

### 4.3 POST-PROCESSING APPROACH

In applications, we first use the DR (or OR, IPS) estimators in Section 4.2 and plug them into the last two constraints of the optimization problem in Section 4.1. As discussed in Section 4.1, if the feasible domain is the empty set, or if there exists $x \in \mathcal{X}$ such that the interval of values of $\tilde{w}(x)$ does not contain 0, then the decision should be considered to violate principal counterfactual fairness. Inspired by (Mishler et al., 2021), we further propose a post-processing method to adjust previously unfair decisions by minimal individual decision changes so that it no longer violates optimization-based fairness evaluation in Section 4.1. The advantage of the post-processing approach is the applicability to any already-in-used models but evaluated unfair (Lohia et al., 2019).

Specifically, consider a set of non-negative parameters $\epsilon(x) = \{\epsilon_{00}(x), \epsilon_{01}(x), \epsilon_{10}(x), \epsilon_{11}(x)\}$ for all $x \in \mathcal{X}$, where each parameter $\epsilon_{ad}(x)$ denotes the probability of forcing the decision $D = d$ on the individuals with $A = a$ and $X = x$. With loss of generality, $\epsilon_{ad}(x) + \epsilon_{a(1-d)}(x) \leq 1$, and let $D'$ be the final decision after the post-processing, then we have

$$\mathbb{P}_\epsilon(D'(a) = d, Y(a) = y \mid X = x) = \mathbb{P}_\epsilon(D' = d, Y = y \mid A = a, X = x)$$
$$= \epsilon_{ad}(x) \cdot \mathbb{P}(Y = y \mid A = a, X = x) + (1 - \epsilon_{a0}(x) - \epsilon_{a1}(x)) \cdot \mathbb{P}(D = d, Y = y \mid A = a, X = x)$$
$$= \epsilon_{ad}(x) \cdot \mathbb{P}(Y(a) = y \mid X = x) + (1 - \epsilon_{a0}(x) - \epsilon_{a1}(x)) \cdot \mathbb{P}(D(a) = d, Y(a) = y \mid X = x)$$
$$= \epsilon_{ad}(x) \cdot \mathbb{P}(D(a) = 1 - d, Y(a) = y \mid X = x) + (1 - \epsilon_{a(1-d)}(x)) \cdot \mathbb{P}(D(a) = d, Y(a) = y \mid X = x)$$

In order to obtain a fair decision $D'$ while minimally changing the original decision $D$, we obtain the estimated $\hat{\epsilon}(x)$ for all $x \in \mathcal{X}$ by solving the following optimization problem.

$$\epsilon^* = \arg\min_\epsilon \frac{1}{n} \sum_{i=1}^n \epsilon_{A_i 0}(X_i) + \epsilon_{A_i 1}(X_i),$$

$$\text{s.t.} \quad w_{0,1,0,0}(x) = w_{1,0,0,0}(x) = w_{0,1,1,1}(x) = w_{1,0,1,1}(x) = 0,$$
$$w_{d_0,d_1,y_0,y_1}(x) \geq 0 \quad \text{for all} \quad d_0, d_1, y_0, y_1 \in \{0, 1\}$$
$$\epsilon_{ad}(x)\epsilon_{a(1-d)}(x) \geq 0 \quad \text{and} \quad \epsilon_{ad}(x) + \epsilon_{a(1-d)}(x) \leq 1 \quad \text{for all} \quad a, d \in \{0, 1\}$$
$$\sum_{a,b} w_{d_0,a,y_0,b}(x) = \mathbb{P}_\epsilon(D'(0) = d_0, Y(0) = y_0 \mid X = x) \quad \text{for all} \quad d_0, y_0 \in \{0, 1\}$$
$$\sum_{a,b} w_{a,d_1,b,y_1}(x) = \mathbb{P}_\epsilon(D'(1) = d_1, Y(1) = y_1 \mid X = x) \quad \text{for all} \quad d_1, y_1 \in \{0, 1\},$$

for all $x \in \mathcal{X}$. In practice, we use the DR estimate in Section 4.2 and then plug it in to the last two constraints to obtain the DR estimates for $\mathbb{P}_\epsilon(D'(a) = d, Y(a) = y \mid X = x)$, and estimate $\hat{\epsilon}(x)$ for all $x \in \mathcal{X}$. Theorem 3 proves the consistency results of the estimated $\hat{\epsilon}(x)$ to the optimal $\epsilon^*(x)$.

**Theorem 3.** *Suppose that* $||\hat{\pi}_a(x) - \pi_a(x)||_2 \cdot ||\hat{\mu}_a^{d,y}(x) - \mu_a^{d,y}(x)||_2 = o_{\mathbb{P}}(n^{-1/2})$ *for all* $x \in \mathcal{X}$ *and $a$ attainable by $A$, then* $||\hat{\epsilon}(x) - \epsilon^*(x)|| = O_P(1/\sqrt{n})$.

### 4.4 THEORETICAL ANALYSIS

After obtaining the optimization solution $\hat{\epsilon}(x)$ in Section 4.3, the adjusted decision function $D'$ is determined by combining original decision $D$ and $\hat{\epsilon}(x)$. Next, we prove that the adjusted decision $D'$

Table 3: Synthetic experiment results for varying models and estimators. The intervals under $w_{0100}$, $w_{1000}$, $w_{0111}$ and $w_{1011}$ show the minimum and maximum values when the other three are set to 0.

| Method | $w_{0100}$ | $\epsilon_{00}$ | $w_{1000}$ | $\epsilon_{10}$ | $w_{0111}$ | $\epsilon_{01}$ | $w_{1011}$ | $\epsilon_{11}$ | CF ↑ | PCF ↑ |
|---|---|---|---|---|---|---|---|---|---|---|
| LR + OR | [-0.636, -0.083] | 0 | [-0.774, -0.083] | 0 | [0.083, 0.223] | 0 | [-0.672, -0.083] | **0.109** | +3.60% | +6.68% |
| LR + IPS | [0.153, 0.231] | 0 | [-0.772, -0.153] | 0 | [-0.666, -0.153] | **0.166** | [-0.715, -0.153] | 0 | +4.71% | +6.89% |
| LR + DR | [-0.631, -0.024] | 0 | [-0.710, -0.024] | 0 | [0.024, 0.196] | 0 | [-0.683, -0.024] | **0.127** | +2.97% | +6.09% |
| SVM + OR | [0.104, 0.260] | 0 | [-0.741, -0.104] | **0.124** | [-0.743, -0.104] | 0 | [-0.619, -0.104] | 0 | +1.47% | +5.15% |
| SVM + IPS | [0.192, 0.227] | 0 | [-0.720, -0.192] | 0 | [-0.739, -0.192] | **0.199** | [-0.732, -0.192] | 0 | +4.56% | +8.98% |
| SVM + DR | [-0.662, -0.034] | 0 | [-0.763, -0.034] | 0 | [0.034, 0.182] | **0.186** | [-0.607, -0.034] | 0 | +5.77% | +8.30% |
| RF + OR | [-0.706, -0.110] | **0.120** | [-0.713, -0.110] | 0 | [-0.689, -0.110] | 0 | [0.110, 0.195] | 0 | +2.18% | +6.67% |
| RF + IPS | [0.163, 0.211] | 0 | [-0.727, -0.163] | 0 | [-0.680, -0.163] | **0.171** | [-0.755, -0.163] | 0 | +6.42% | +8.35% |
| RF + DR | [-0.713, -0.051] | 0 | [-0.676, -0.051] | 0 | [0.051, 0.253] | **0.203** | [-0.661, -0.051] | 0 | +6.49% | +9.02% |
| NB + OR | [-0.674, -0.161] | 0 | [-0.744, -0.161] | 0 | [0.161, 0.232] | **0.173** | [-0.742, -0.161] | 0 | +5.04% | +8.31% |
| NB + IPS | [-0.821, -0.175] | 0 | [0.175, 0.192] | 0 | [-0.775, -0.175] | 0 | [-0.577, -0.175] | **0.181** | +8.88% | +9.31% |
| NB + DR | [-0.793, -0.189] | 0 | [-0.688, -0.189] | 0 | [-0.707, -0.189] | 0 | [0.189, 0.208] | **0.192** | +4.83% | +9.16% |

tends to be fair as sample size $n \to \infty$. To this end, consider the following programming problem

$$\alpha^* = \arg\min_w \frac{1}{n} \sum_{i=1}^n w_{0,1,0,0}^2(X_i) + w_{1,0,0,0}^2(X_i) + w_{0,1,1,1}^2(X_i) + w_{1,0,1,1}^2(X_i),$$

$$\text{s.t.} \quad w_{d_0,d_1,y_0,y_1}(x) \geq 0 \quad \text{for all} \quad d_0, d_1, y_0, y_1 \in \{0,1\},$$

$$\sum_{a,b} w_{d_0,a,y_0,b}(x) = \mathbb{P}_{\hat\epsilon}(D'(0) = d_0, Y(0) = y_0 \mid X = x) \quad \text{for all} \quad d_0, y_0 \in \{0,1\},$$

$$\sum_{a,b} w_{a,d_1,b,y_1}(x) = \mathbb{P}_{\hat\epsilon}(D'(1) = d_1, Y(1) = y_1 \mid X = x) \quad \text{for all} \quad d_1, y_1 \in \{0,1\},$$

for all $x \in \mathcal{X}$, where $\mathbb{P}_{\hat\epsilon}(D'(a), Y(a) \mid X = x)$ is the joint distribution of the potential outcomes $(D'(a), Y(a))$ for the post-processed decision function $D'$ using $\hat\epsilon(x)$ obtained in Section 4.3. We use the mean sum of squares as the metric for evaluating principal counterfactual fairness. Theorem 4 proves the consistency results of $\alpha^*$ to 0, validating the effectiveness of the post-processing approach.

**Theorem 4.** *Suppose that* $||\hat\pi_a(x) - \pi_a(x)||_2 \cdot ||\hat\mu_a^{d,y}(x) - \mu_a^{d,y}(x)||_2 = o_{\mathbb{P}}(n^{-1/2})$ *for all* $x \in \mathcal{X}$ *and $a$ attainable by $A$, then* $||\alpha^*||_1 = O_P(1/\sqrt{n})$.

## 5 EMPIRICAL INVESTIGATION

To verify the effectiveness of the post-processing approach, we conduct experiments on both synthetic and real-world dataset. The performance is evaluated by two metrics: counterfactual fairness (CF): $\mathbb{P}(D(0) = D(1))$ and principal counterfactual fairness (PCF): $\mathbb{P}(D(0) = D(1) \mid Y(0) = Y(1))$. For all experiments, we calculate the values of CF and PCF before and after the post-processing operation and report the percentage change of each metric respectively.

### 5.1 SYNTHETIC EXPERIMENT

Synthetic data are generated from a structural equation model based on a random DAG with 10 nodes and 40 directed edges according to the Erdős-Rényi (ER) model, where four different models: Logistic Regression (LR), Support Vector Machine (SVM), Random Forest (RF) and Naive Bayes (NB) respectively to obtain the estimations of $\mathbb{P}(Y = 1 \mid A = a, X = x)$ as the decisions $D(a, x)$ (see Appendix C for details). Then we check whether the optimization equation in Section 4.1 is solvable. If the feasible domain is empty, we further use the post-processing method in Section 4.3 to obtain $\hat\epsilon_{ad}(x)$. Table 3 shows the synthetic experiment results. First, the intervals of the four $w$ do not contain 0, implying that the optimization equation has no solution. At this point there will be one $\hat\epsilon_{ad}(x)$ nonzero, while the other three $\hat\epsilon_{ad}(x)$ are 0. Second, after the flip based on the $\hat\epsilon_{ad}(x)$, there are positive changes in both PCF and CF, and the increase in PCF is more pronounced than CF for all models. This is because our approach focuses only on the population with $Y(0) = Y(1)$.

### 5.2 REAL-WORLD EXPERIMENT

The STUDENTINFO file in the Open University Learning Analytics Dataset (OULAD) dataset (Kuzilek et al., 2017) is used for the real-world experiment. The data attributes include demographic informa-

Table 4: Real-world experiment results for different subgroups. The intervals under $w_{0100}$, $w_{1000}$, $w_{0111}$ and $w_{1011}$ show the minimum and maximum values when the other three $w$ are 0.

| Subgroup | $w_{0100}(X)$ | $\epsilon_{00}(X)$ | $w_{1000}(X)$ | $\epsilon_{10}(X)$ | $w_{0111}(X)$ | $\epsilon_{01}(X)$ | $w_{1011}(X)$ | $\epsilon_{11}(X)$ | CF ↑ | PCF ↑ |
|---|---|---|---|---|---|---|---|---|---|---|
| None | [-0.716, 0.024] | 0 | [-0.747, 0.020] | 0 | [-0.457, 0.078] | 0 | [-0.078, 0.268] | 0 | - | - |
| $X_1 \geq 120$ | [-0.131, 0.274] | 0 | [-0.713, 0.059] | 0 | [-0.778, 0.093] | 0 | [-0.377, 0.131] | 0 | - | - |
| $X_1 < 120$ | [-0.580, -0.030] | 0 | [0.030, 0.239] | 0 | [-0.747, -0.030] | **0.040** | [-0.702, -0.030] | 0 | +1.35% | +1.79% |
| $X_2 > 0$ | [0.039, 0.242] | 0 | [-0.675, -0.039] | **0.049** | [-0.658, -0.039] | 0 | [-0.705, -0.039] | 0 | +3.52% | +3.97% |
| $X_2 = 0$ | [-0.715, 0.007] | 0 | [-0.575, 0.007] | 0 | [-0.332, 0.092] | 0 | [-0.376, 0.213] | 0 | - | - |
| $X_1 \geq 120, X_2 > 0$ | [0.173, 0.287] | 0 | [-0.754, -0.173] | 0 | [-0.715, -0.173] | **0.196** | [-0.704, -0.173] | 0 | +3.80% | +8.51% |
| $X_1 \geq 120, X_2 = 0$ | [-0.749, 0.000] | 0 | [-0.444, 0.057] | 0 | [-0.748, 0.001] | 0 | [-0.057, 0.246] | 0 | - | - |
| $X_1 < 120, X_2 > 0$ | [0.176, 0.220] | 0 | [-0.738, -0.176] | 0 | [-0.731, -0.176] | **0.184** | [-0.706, -0.176] | 0 | +5.22% | +9.59% |
| $X_1 < 120, X_2 = 0$ | [-0.822, -0.231] | 0 | [-0.666, -0.231] | 0 | [-0.742, -0.231] | 0 | [0.231, 0.304] | **0.124** | +3.10% | +5.02% |

Note: In real-world experiments, $X_1 = studied\_credits$ and $X_2 = num\_of\_prev\_attempts$.

tion about the students such as gender, age, education level, disability, and other attributes as well as their final grades. This dataset contains 32,593 students and 11 attributes. We treat disability as the sensitive attribute and binarize the final grades as the outcome of interest. First we learn a CPDAG from the raw data using the PC algorithm in the causal-learn package. We find *studied_credits*: the total number of credits for the modules the student is currently studying (denoted as $X_1$) and *num_of_prev_attempts*: the number of how many times the student has attempted this module (denoted as $X_2$) with an undirected edge between it and the disability. Therefore we sample four DAGs from the learned CPDAG corresponding to the four cases of no subgroup, $X_1$ as subgroups, $X_2$ as subgroups, and both $X_1$ and $X_2$ as subgroups, respectively. For each DAG, we determine the path coefficients based on linear regression and treat the residual of the regression as noise. For each subgroup, the subsequent steps are the same as in the simulation experiments. The LR model is used to obtain the decision $D$ and the DR estimator is used to estimate $\mathbb{P}(D(a) = d, Y(a) = y \mid X = x)$.

Table 4 shows the real-world experiment results. When solving the optimization problem in Section 4.1 on the whole population, we find that the interval of all four $w$ covers 0, i.e., the current algorithm already satisfies the principle of counterfactual fairness. Therefore, post-processing is unnecessary at this point, so there is no corresponding change in CF and PCF. When using $X_1$ to divide the population, the optimization equation for the subgroup of $X_1 \geq 120$ has no solution, and when grouping according to $X_2$, the optimization equation for the subgroup of $X_2 > 0$ has no solution. When we divide the whole population into four subgroups, we find that the optimization equations for three of subgroups are unsolvable. Compared to the case of two subgroups, for the unsolvable subgroups, the distance between zero and the interval of the four $w$ and the value of the non-zero $\hat{\epsilon}_{ad}(x)$ is significantly larger when the population is divided into four subgroups. This indicates that the constraint is violated to a stronger extent when the population is divided into more subgroups. Meanwhile, for the case of four subgroups, the CF and PCF of each unsolvable subgroup change more due to the larger $\hat{\epsilon}_{ad}(x)$. In addition, for solvable subgroups, the interval of $w_{0100}$ and $w_{0111}$ are very close to exclude 0 when the population is divided into four subgroups, which further indicates that as the number of population group increases, each subgroup becomes more difficult to satisfy the optimization equation. Finally, the growth of PCF is larger than that of CF for all unsolvable subgroups, which shows that our approach is more effective on the population with $Y(0) = Y(1)$.

## 6 CONCLUSION

This paper studies the question of *"which attributes and individuals should be protected"* in the context of counterfactual fairness. Motivated by the example that disability serves as a sensitive attribute for different outcomes of interest (e.g., college admissions, athlete selections), we suggest that when and how to enforce fairness is expected to depend on whether the protected attribute has no individual causal effect on the outcome of interest. Formally, we propose *principal counterfactual fairness*, and theoretically derive the necessary conditions for an algorithm to satisfy principal counterfactual fairness based on statistical bounds. Based on this, we further propose a principled post-processing approach to achieve principal counterfactual fairness with minimal individual decision changes. A limitation of this work is that the principal counterfactual fairness is partially identified, i.e., we cannot give unbiased point estimates from the data, but can give its statistical bounds and a falsification method. We leave it to future work about how to develop new identification and estimation strategies under practical assumptions. In addition, combining causal discovery to achieve decisions that satisfy the principal counterfactual fairness also serves as an interesting future topic.

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

## A   PROOFS

**Proof of Theorem 2.** In this proof, for the simplicity of exposition, we only consider a special case that the conditional set is empty

Let

$$\varphi_{a,d,y}(A, X, D, Y; \pi, \mu) = \mu(X) + \{\mathbb{I}(A = a) \cdot \mathbb{I}(D = d, Y = y) - \mu(X)\}/\pi(X)$$

And

$$\hat{\mathbb{P}}^{DR}(D(a) = d, Y(a) = y) - \mathbb{P}(D(a) = d, Y(a) = y) = A_{1n} + A_{2n}$$

$$A_{1n} = \frac{1}{n} \sum_{i=1}^{n} [\varphi_{a,d,y}(A, X, D, Y; \pi, \mu) - \mathbb{P}(D(a) = d, Y(a) = y)],$$

$$A_{2n} = \frac{1}{n} \sum_{i=1}^{n} [\varphi_{a,d,y}(A, X, D, Y; \hat{\pi}, \hat{\mu}) - \varphi_{a,d,y}(A, X, D, Y; \pi, \mu)].$$

By the central limit theorem, $\sqrt{n} A_{1n}$ is asymptotic normal. For $A_{1n}$, Define the Gateaux derivative of the generic function $g$ in the direction $[\hat{\pi} - \pi, \hat{\mu} - \mu]$ by $\partial_{[\hat{\pi}-\pi,\hat{\mu}-\mu]}g$. By a Taylor expansion for $\mathbb{E}[A_{2n}]$ yields that

$$\mathbb{E}[A_{2n}] = \mathbb{E}[\varphi_{a,d,y}(A, X, D, Y; \hat{\pi}_a, \hat{\mu}_a^{d,y}) - \varphi_{a,d,y}(A, X, D, Y; \pi_a, \mu_a^{d,y})]$$

$$= \partial_{[\hat{\pi}_a-\pi_a,\hat{\mu}_a^{d,y}-\mu_a^{d,y}]}\mathbb{E}[\varphi_{a,d,y}(A, X, D, Y; \pi_a, \mu_a^{d,y})]$$

$$+ \frac{1}{2}\partial^2_{[\hat{\pi}_a-\pi_a,\hat{\mu}_a^{d,y}-\mu_a^{d,y}]}\mathbb{E}[\varphi_{a,d,y}(A, X, D, Y; \pi_a, \mu_a^{d,y})] + \cdots$$

The first-order term

$$\partial_{[\hat{\pi}_a-\pi_a,\hat{\mu}_a^{d,y}-\mu_a^{d,y}]}\mathbb{E}[\varphi_{a,d,y}(A, X, D, Y; \pi_a, \mu_a^{d,y})] = 0,$$

For the second-order term, we get

$$\frac{1}{2}\partial^2_{[\hat{\pi}_a-\pi_a,\hat{\mu}_a^{d,y}-\mu_a^{d,y}]}\mathbb{E}[\varphi_{a,d,y}(A, X, D, Y; \pi_a, \mu_a^{d,y})]$$

$$= \mathbb{E}\left[\frac{I(A = a)\{I(D = d, Y = y) - \mu_a^{d,y}(X)\}}{(\pi_a(X))^3}\{\hat{\pi}_a(X) - \pi_a(X)\}^2\right.$$

$$\left. + \frac{I(A = a)}{(\pi_a(X))^2}\{\hat{\pi}_a(X) - \pi_a(X)\}\{\hat{\mu}_a^{d,y}(X) - \mu_a^{d,y}(X)\}\right]$$

$$= \mathbb{E}\left[\frac{1}{(\pi_a(X))}\{\hat{\pi}_a(X) - \pi_a(X)\}\{\hat{\mu}_a^{d,y}(X) - \mu_a^{d,y}(X)\}\right]$$

$$\leq C \cdot ||\hat{\pi}_a(X) - \pi_a(X)||_2 \cdot ||\hat{\mu}_a^{d,y}(X) - \mu_a^{d,y}(X)||_2$$

$$= o_P(n^{-1/2}),$$

where $C$ is a finite constant. All higher-order terms can be shown to be dominated by the second-order term. Therefore,

$$\mathbb{E}[A_{2n}] = o_P(n^{-1/2}).$$

This completes the proof.

$\square$

We first introduce a lemma from (Shapiro, 1991), and use it in the proof of Theorem 2 and 3.

**Lemma 2** Let $\Theta$ be a compact subset of $\mathbb{R}^k$. Let $C(\Theta)$ denote the set of continuous real-valued functions on $\Theta$, with $\mathcal{L} = C(\Theta) \times \ldots \times C(\Theta)$ the $r$-dimensional Cartesian product. Let $\psi(\theta) = (\psi_0, \ldots, \psi_r) \in \mathcal{L}$ be a vector of convex functions. Consider the quantity $\alpha^*$ defined as the solution to the following convex optimization program:

$$\alpha^* = \min_{\theta \in \Theta} \psi_0(\theta)$$

$$\text{subject to } \psi_j(\theta) \leq 0, j = 1, \ldots, r$$

Assume that Slater's condition holds, so that there is some $\theta \in \Theta$ for which the inequalities are satisfied and non-affine inequalities are strictly satisfied, i.e. $\psi_j(\theta) < 0$ if $\psi_j$ is non-affine. Now consider a sequence of approximating programs, for $n = 1, 2, \ldots$:

$$\widehat{\alpha}_n = \min_{\theta \in \Theta} \quad \widehat{\psi}_{0n}(\theta)$$

$$\text{subject to } \widehat{\psi}_{jn}(\theta) \leq 0, j = 1, \ldots, r$$

with $\widehat{\psi}_n(\theta) := \left( \widehat{\psi}_{0n}, \ldots, \widehat{\psi}_{rn} \right) \in \mathcal{L}$. Assume that $f(n)(\widehat{\psi}_n - \psi)$ converges in distribution to a random element $W \in \mathcal{L}$ for some real-valued function $f(n)$. Then:

$$f(n) \left( \widehat{\alpha}_n - \alpha_0 \right) \rightsquigarrow L$$

for a particular random variable $L$. It follows that $\widehat{\alpha}_n - \alpha_0 = O_{\mathbb{P}}(1/f(n))$.  $\square$

**Proof of Theorem 3.** According to **Theorem** 2, the constraint part of the empirical optimization problem is asymptotic to the constraint part of the oracle optimization problem. This can be proved directly from **Lemma** 2.  $\square$

**Proof of Theorem 4.** Because $\epsilon^*$ is solved by the programing question, it means,

$$\min_{w} \frac{1}{n} \sum_{i=1}^{n} w_{0,1,0,0}^2(X_i) + w_{1,0,0,0}^2(X_i) + w_{0,1,1,1}^2(X_i) + w_{1,0,1,1}^2(X_i) = 0,$$

$$\text{s.t.} \quad w_{d_0,d_1,y_0,y_1}(x) \geq 0 \quad \text{for all} \quad d_0, d_1, y_0, y_1 \in \{0, 1\}$$

$$\sum_{a,b} w_{d_0,a,y_0,b}(x) = \mathbb{P}_{\epsilon^*}(D'(0) = d_0, Y(0) = y_0 \mid X = x) \quad \text{for all} \quad d_0, y_0 \in \{0, 1\}$$

$$\sum_{a,b} w_{a,d_1,b,y_1}(x) = \mathbb{P}_{\epsilon^*}(D'(1) = d_1, Y(1) = y_1 \mid X = x) \quad \text{for all} \quad d_1, y_1 \in \{0, 1\},$$

In the same way, we can prove that, for the following programing,

$$\alpha^* = \min_{w} \frac{1}{n} \sum_{i=1}^{n} w_{0,1,0,0}^2(X_i) + w_{1,0,0,0}^2(X_i) + w_{0,1,1,1}^2(X_i) + w_{1,0,1,1}^2(X_i),$$

$$\text{s.t.} \quad w_{d_0,d_1,y_0,y_1}(x) \geq 0 \quad \text{for all} \quad d_0, d_1, y_0, y_1 \in \{0, 1\}$$

$$\sum_{a,b} w_{d_0,a,y_0,b}(x) = \mathbb{P}_{\hat{\epsilon}}(D'(0) = d_0, Y(0) = y_0 \mid X = x) \quad \text{for all} \quad d_0, y_0 \in \{0, 1\}$$

$$\sum_{a,b} w_{a,d_1,b,y_1}(x) = \mathbb{P}_{\hat{\epsilon}}(D'(1) = d_1, Y(1) = y_1 \mid X = x) \quad \text{for all} \quad d_1, y_1 \in \{0, 1\},$$

According to the proof of **Theorem** 2. $\sqrt{n}\{\hat{\epsilon} - \epsilon\} \rightsquigarrow W$, $W$ is a random element, the constraint part convergence at rate $\sqrt{n}$. Using **Lemma** 2, $\sqrt{n}\alpha^*$ converges to a random element.  $\square$

## B  ESTIMATION OF NUISANCE PARAMETERS WITH SAMPLE SPLITTING IN THEOREM 2

If both $\hat{\pi}_a(X)$ and $\mu_a^{d,y}$ are not assumed to be sufficiently "well-behaved," i.e. if they do not belong to Donsker classes, we need to estimate them by a data-splitting method (Chernozhukov et al., 2018; Mishler et al., 2021; wager and Athey, 2018).

Let $K$ be a small positive integer, and (for simplicity) suppose that $m = n/K$ is also an integer. Let $I_1, \ldots, I_K$ be a random partition of the index set $I = \{1, \ldots, n\}$ so that $\#I_k = m$ for $k = 1, \ldots, K$. Denote $I_k^C$ as the complement of $I_k$.

**Step 1.** Nuisance parameter training for each sub-sample.

**for** $k = 1$ **to** $K$ **do**

(1) Construct estimates $\tilde{\pi}_a(x)$ and $\tilde{\mu}_a^{d,y}(X)$ using the sample with $I_k^C$.

(2) Obtain the predicted values of $\hat{\pi}_a(X)$, and $\hat{\mu}_a^{d,y}(X)$ for $i \in I_k$.

**end**

**Step 2.** All the predicted values $\tilde{\pi}_a(X_i)$ and $\tilde{\mu}_a^{d,y}(X_i)$ for $i \in I$ consist of the estimates of $\pi_a(X_i)$ and $\mu_a^{d,y}(X_i)$, denoted as $\hat{\pi}_a(X_i)$, and $\hat{\mu}_a^{d,y}(X_i)$, respectively.

## C  EXPERIMENTAL DETAILS OF SYNTHETIC EXPERIMENT

We first randomly generate a DAG with 10 nodes and 40 directed edges according to the Erdős-Rényi (ER) model. Following the previous study (Zuo et al., 2022), the path coefficients $\beta_{jk}$ of directed edges $X_j \rightarrow X_k$ are sampled from a Uniform$([-2, -0.5] \cup [0.5, 2])$ distribution. The data are generated according to the following equation:

$$X_k = \sum_{X_j \in pa(X_k)} \beta_{jk} X_j + \epsilon_i, i = 1, \ldots, n,$$

where $pa(X_k)$ represents the parent nodes of $X_k$, noise $\epsilon_i \sim N(0, 2.5)$ and $n$ is the sample size, which is 1,000 in the experiment. Then we randomly select two nodes as the outcome $Y$ and the sensitive attribute $A$, respectively. $A$ is drawn from a Binomial([0,1]) with probability $\sigma(\sum_{X_j \in pa(A)} \beta_{jA} X_j + \epsilon_i)$, where $\sigma(\cdot)$ denotes the sigmoid function. To evaluate our post-processing approach, we need to have $Y(0), Y(1), D(0)$ and $D(1)$ for each sample. Therefore, we generate $Y_i(a)$ from a Binomial([0,1]) with probability $\sigma(\sum_{X_j \in pa(Y)} \beta_{jY} X_j(a) + \epsilon_i)$, where $X(a)$ means the value of $X$ when $A = a$, and use four different models: Logistic Regression (LR), Support Vector Machine (SVM), Random Forest (RF) and Naive Bayes (NB) respectively to obtain the estimations of $\mathbb{P}(Y = 1 \mid A = a, X = x)$ as the decisions $D(a, x)$. After obtaining potential outcomes for each individual, we estimate $\mathbb{P}(D(a) = d, Y(a) = y \mid X = x)$ using OR, IPS and DR estimators, respectively, where $d$ and $y \in \{0, 1\}$. Then we check whether the optimization equation in Section 4.1 is solvable. If the feasible domain is empty, we further use the post-processing method in Section 4.3 to obtain $\hat{\epsilon}_{ad}(x)$. Finally, we flip $D(0)$ and $D(1)$ using $\hat{\epsilon}_{ad}(x)$ as the corresponding probabilities and compute the percentage change of CF and PCF.

