# OpenReview forum: "Rethinking Counterfactual Fairness: On Which Individuals to Enforce, and How?"
_ICLR.cc/2024/Conference — Submitted to ICLR 2024_

### Official Review · Reviewer_wXfm · 2023-10-30

**Soundness:** 2 fair
**Presentation:** 2 fair
**Contribution:** 1 poor
**Rating:** 3
**Confidence:** 5

**Summary:**

This paper proposes a new fairness definition based on counterfactual fairness (CF). Based on the potential outcomes framework, it proposes principal counterfactual fairness (PCF) where it conditions CF on the joint distribution of the potential outcomes of individuals, treating membership to the protected group $A$ as the treatment. This new definition is based on the concept of principal stratification from the causal inference literature. PCF looks at whether an algorithm is counterfactually fair for individuals that would have had the same decision outcome regardless of $A$. In other words, PCF looks at individuals for whom $A$ has no individual causal effect on the decision outcome. In doing so, according to the paper, PCF addresses the shortcomings of CF in answering which attributes should be protected and why.

The paper proposes an optimization-based approach for testing PCF, including theoretical guarantees, and further proposes a post-processing approach to achieve PCF. It evaluates the proposed methods on synthetic and real data, comparing PCF only to CF.

**Strengths:**

S1: The general theme of extending CF to account for the (potential) outcomes is interesting and understudied in causal fair ML. The use of principal stratification is novel and worth exploring further. Using such information to explore sub-groups, or strata, of the population seems promising for linking (more explicitly) predictions and outcomes under CF.

S2: PCF, conceptually, seems to raise a potential tension between causal effects at a population level versus at an individual level, meaning to what extent is (or should be) the causal effect homogenous over all individuals in the population.

S3: Table 1 is nice summary on fairness definitions and links to the outcome $Y$ and decision $D$.

**Weaknesses:**

W1: It’s unclear to me where this “shortcoming” of CF (and extensions) on defining which attributes to enforce comes from. Overall, the paper is not well situated within the relevant causal fairness literature.

First, Kusner et al. (2017) never claim to tackle this problem and rather focus on how to achieve counterfactual fairness given auxiliary causal knowledge. I’d argue the same for all subsequent works like Chiappa (2019). The problem of “which attributes should be protected” has always been presented as either context specific or historically motivated or both.

Second, the paper ignores existing literature that has looked at how CF treats protected attributes and why this is an issue (like “What’s sex got to do with fair machine learning?” by Hu and Kohler-Haussman (2020) or the works on algorithmic recourse and algorithmic discrimination via counterfactual reasoning that appear often in fairness conferences). It’s, thus, hard to see the novelty and impact of PCF relative to existing works on CF and causal fairness.

W2: CF is a fairness notion based on structural causal models (SCM), yet this paper doesn’t introduce SCM or discuss it beyond the second footnote. It takes for granted the link between SCM and PO, which is not a common view regarding counterfactual generation (e.g., under Pearl’s causal hierarchy). In fact, as far as I understand, these two frameworks coincide only when under a randomized control trail setting, which is not the case in the experimental section. It’s, thus, difficult to link PCF to CF itself and to further compare these two notions. It would help the paper if CF was presented through SCM.

W3: The motivation is unclear and, thus, hard to follow the overall usefulness of PCF and all the definitions proposed. The admissions example looks like two separate decision processes: school performance versus athletic performance. How are these related to the decision algorithm?

**Questions:**

Q1: What happens when the protected attribute does have a causal effect on the outcome of interest? Are we not able to say anything or do something about the fairness problem in that case?

Q2 (from W1): Why should it be up to CF to define on which attributes to apply the method? Overall, I don’t understand where this shortcoming comes from as it is not a view shared by the field (to the best of my knowledge) nor is there work cited by the authors referring to such view.

Q3 (from W3): One thing is to enter university and another one is to enter under an athletic scholarship: these are separate (or at least sequential) decision contexts, no? This motivating example is confusing. Could the authors elaborate further or provide a real life example that illustrates why PCF is better than CF?

---

### Official Review · Reviewer_tVGj · 2023-10-31

**Soundness:** 2 fair
**Presentation:** 2 fair
**Contribution:** 3 good
**Rating:** 6
**Confidence:** 4

**Summary:**

In this manuscript, the authors propose the notion of *principal counterfactual fairness*, which represents a relaxed version of the previously proposed notion of *counterfactual fairness* (Kusner et al., 2017). Whereas the latter fairness definition posits that taking a counterfactual with respect to a sensitive attribute may *never* change the algorithm's prediction, the newly proposed definition applies this constraint only to those samples for which taking such a counterfactual also does not result in a change of the target label.

As a motivating example for this new definition, the authors present the case of physical disabilities, which would not be expected to affect a student's college performance and should thus also not affect a model's predictions. In the case of athlete selection, however, such disabilities would likely affect outcomes, and thus, following the newly proposed fairness definitions, the disability attribute would be allowed to influence model predictions.

Technically, the authors present necessary conditions for the new fairness definition to hold, as well as a postprocessing approach that can change a model's predictions in order to satisfy these necessary conditions. The approach can be implemented by solving a constrained quadratic programming problem, where some of the constraints are determined from the observed data using a doubly robust estimator.

The proposed postprocessing method is evaluated on a synthetic dataset as well as a real-world standard student success dataset. The empirical results indicate that the proposed method successfully improves the path-specific counterfactual fairness of the model (even though it only enforces necessary, not sufficient, conditions).

**Strengths:**

The authors address the important question of with respect to which sensitive attributes a model should be (counterfactually) invariant. They meticulously position their new definition in the context of previously proposed ones, clearly spelling out the differences between the different definitions.

The proposed notion of principal counterfactual fairness is, at least to my knowledge, new, and I have also not seen very similar notions discussed in the literature. It adds an interesting new perspective to discussions of the different notions of algorithmic fairness that have been discussed widely, and I found the paper quite useful in this regard.

Despite the fundamental difficulty of enforcing the new definition directly (due to lack of observability), the authors develop a practical approach for at least getting closer to satisfying it.

I mostly found the paper to be well written.

**Weaknesses:**

## Normative assumptions and implications
The paper is very explicitly framed in terms of a normative question: *which attributes and individuals should be protected*? Given this highly normative framing (note the *should*), the paper sorely lacks a clear discussion of the normative assumptions underlying the fairness definition adopted by the authors, as well as its implications.

The authors suggest that "when and how to enforce fairness is expected to depend on whether the protected attribute has no individual causal effect on the outcome of interest," and they more specifically posit that (sensitive) attributes that *do* influence outcomes are "allowed" to also influence model predictions. However, while probably appropriate in some scenarios, this will most certainly not always be considered acceptable. One might - and we as a society do - very well decide that certain attributes are not allowed to influence decisions *even though they will probably affect the outcome of interest*. As one of many examples, consider the case of pregnancy/childbirth. In many countries, we do not allow firing pregnant women (or taking the fact whether a woman plans to become pregnant into account in hiring situations), even though them being pregnant - and thus not working for a while - will most certainly affect some outcome measure of interest to the company. Under the definition proposed by the authors, preferentially firing pregnant women would be deemed "fair".

Moreover, the authors' proposed fairness notion does not differentiate between fair and unfair paths (Chiappa, 2019). There is an important difference between the situation in which a sensitive attribute *directly* affects outcomes of interest, and the situation in which it affects outcomes via some other intermediate factors. For instance, a physical disability might not influence exam grades directly, but it might very well do so indirectly via factors such as social discrimination, access barriers of all sorts, and a generally heightened difficulty of participating in "normal" student life. This distinction between direct and indirect effects seems crucial to me in many other settings as well, such as racial or gender-based discrimination. However, the authors' definition 7 only asks whether taking the sensitive attribute counterfactual will change the outcome for this individual, not *why* the outcome changes (or not). (More technically, whenever the authors write about "individual causal effects", they seem to refer to the *total* individual causal effect, not the *direct* individual causal effect. This might be worth spelling out explicitly.)

Many of the results in the paper also rely on the validity of Assumption 1 (Ignorability), but the implications of this assumption are not clearly discussed. The assumption is that $A \perp (Y(1), Y(0), D(1), D(0)) \mid X$. I believe that this rules out, for instance, the case in which $A$ has a direct causal influence on $Y$ and $A$ is not fully captured by $X$? Any direct discrimination setting would fall under this category. A careful discussion of the situations in which this assumption may or may not be justified would be important.

Finally, the authors write in various places that their proposed definition "degenerates" to standard counterfactual fairness under certain conditions. This implies that the definition proposed by the authors (enforcing fairness only on individuals on which the sensitive attribute does not affect the outcome) were strictly superior to the definition which enforces counterfactual fairness on all individuals. The two are simply two different fairness conceptions, however, and the more appropriate one can only be selected based on normative deliberation concerning a specific application at hand.

None of these critiques imply that the fairness definition proposed by the authors is not useful, or that it is never valid. However, proposing a new fairness conception comes with the important obligation to also carefully point out the situations under which this definition may or may not be justified. Currently, the paper reads as if the authors believe to have definitely and conclusively answered the question "which sensitive attributes to enforce fairness on" once and for all, which they most certainly have not.

## Clarity issues
There were a few places throughout the manuscript in which I found it hard to follow the authors' argumentation. This concerned, most importantly, the empirical investigations / case studies. Some things I was wondering about:
- The authors write that "there will be one [epsilon hat] nonzero, while the other three are 0." - why is this the case? Why cannot multiple epsilons be non-zero? It is not obvious to me that the optimization problem on p. 7 (equation numbers would be appreciated, by the way) would always have a solution with at most one non-zero entry? Also, epsilon_hat should depend on x - what are the epsilons displayed in Table 3 and Table 4?
- Percent changes are fine, but the absolute values of CF and PCF should also be shown.
- I found the whole StudentInfo case study very hard to follow. What does it mean that the authors "find studied_credits [...] and num_of_prev_attempts [...] with an undirected edge between it and the disability"? What do the authors mean by "we sample four DAGs from the learned CPDAG corresponding to the four cases of no subgroup, X1 as subgroups, X2 as subgroups, and both X1 and X2 as subgroups"? The abbreviation CPDAG is never spelled out. Also, there was no previous mention of "subgroups" and I am unsure what these refer to in the authors' framework / what the algorithmic / DAG implications of these subgroup choices are. Drawing the assumed DAGs would also be highly appreciated. Finally, can the authors formulate the conclusions of this case study in natural language and in terms of the actual case? Did they/we learn anything about fairness in this specific scenario by conducting the described analysis? The discussion of the results is highly technical and does not discuss any insights gained about the actual case.

Other places that left me slightly confused or took me a while to understand:
- In table 2, it was unclear to me what was actually being shown in the table. (What are the things being shown in the different table cells? What do they mean?)
- It is not clear to me how exactly Definition 5 (Principal counterfactual equalized odds) is in conflict with $P(D(a)=1 \mid Y(a)=Y(a')=1) > P(D(a)=1 \mid Y(a)=Y(a')=0)$? (Also, the sentence "Although the protected attribute has no individual causal effect ..." is missing a main clause.)
- It took me a moment to understand what the authors mean by "each parameter [epsilon] denotes the probability of forcing the decision D=d". I would suggest formally defining this notion, i.e., something like $\epsilon_{ad}(x) := P(D'(a)=d, D(a)=1-d \mid x)$? Also, directly below, it says "With loss of generality" - is this a typo? If not, and generality is actually lost, could the authors elaborate on why this is the case?

**Questions:**

A few minor issues:
- The authors seem to suggest that the extension of their framework to the continuous case would be rather trivial; this seems a little optimistic to me?
- In the introduction, the authors characterize demographic parity as "requiring fairness to hold on all individuals", but demographic parity is a purely statistical notion of group fairness and says nothing about individual fairness.
- In the introduction, the authors write about "the joint potential outcomes of the decision on outcome". I suggest rephrasing this.
- In section 2, I would suggest clearly specifying the inputs to the model $D$. (In particular, is $A$ an input or not?)
- "Principal" is misspelled in various places as "principle".
- In the conclusion, the authors write that they "propose a principled post-processing approach to achieve principal counterfactual fairness" - this seems like an overstatement to me, given that the proposed method only achieves satisfaction of a necessary (but not sufficient) condition for principal counterfactual fairness to hold?

---

### Official Review · Reviewer_CWdU · 2023-10-31

**Soundness:** 2 fair
**Presentation:** 3 good
**Contribution:** 2 fair
**Rating:** 3
**Confidence:** 4

**Summary:**

The paper uses potential outcomes and principal stratification to define some new fairness criteria. Similar in spirit to principal fairness of Imai and Jiang, these definitions say that decision probabilities should be equal conditional on potential outcomes being equal. It provides some formal comparisons with other definitions (like counterfactual fairness), describes implementations using some standard assumptions (ignorability) and results in the potential outcome literature (bounds, estimators), and explores application to some synthetic and real datasets.

**Strengths:**

The paper is fairly clear in its notation, writing, and comparison with other closely related work.

**Weaknesses:**

1) It's not clear to me how the proposed definition relates to the motivating question of "which attributes and which individuals should be protected?" The proposed definitions take the choices about which variables are sensitive attributes, outcomes, and decisions as already given. The closest relation that I can understand to the motivating question is that there are example problems where a certain sensitive attribute (e.g. leg related disability) might not fit practically with a standard fairness definition for a certain kind of decision (e.g. selecting runner athletes), and using the proposed fairness definition would allow decisions that would violate the other fairness definitions.

2) Without some clarification and additional assumptions, the proposed definitions seem very weak to me. They say something like: "**decisions should be fair, but only for individuals whose outcomes are already fair**." This seems trivial to satisfy, and if there are examples with infeasibility I would guess that's because the outcomes in those examples are not already fair. To make the requirement less trivial, I believe it would need to be augmented with assumptions about the causes of unfairness that clarify how it's possible for sensitive attributes to have no effect on the outcome while still resulting in a problem that requires any constraints. Otherwise, why is the given problem even an example involving fairness at all?

3) The relevance to ICLR is not specifically strong, but this is not a key weakness.

**Questions:**

How realistic are the assumptions, like ignorability, in typical fairness applications?

Could some structural, graphical models help clarify any of the definitions or motivating examples?

**Details Of Ethics Concerns:**

I believe the framework proposed here would be particularly amenable to "fairwashing" and that the authors have not addressed this risk sufficiently

---

### Official Review · Reviewer_R7Tu · 2023-11-01

**Soundness:** 2 fair
**Presentation:** 2 fair
**Contribution:** 2 fair
**Rating:** 5
**Confidence:** 3

**Summary:**

The paper proposes a new causal fairness definition called Principal Counterfactual Fairness, which essentially asks for counterfactual fairness on individuals for which the protected attribute has no causal effect on the outcome of interest. They motivate this definition by comparing it to other fairness definitions, show how to claim that the algorithm doesn't satisfy the proposed fairness definition and how to post-process a model to have the model satisfy the fairness, and verify their approach on synthetic + real world datasets.

At a technical level, they show a lower and upper bound on equalized odds value (lemma 1) which can be used to assert when the their fairness definition is violated (Theorem 1) using only observational data. Furthermore, they show an optimization-based estimation strategy to evaluate whether the proposed fairness definition is violated (Section 4.1-4.2). Finally, they show how to post-process a given model to enforce fairness.

Lastly, they evaluate their approach on synthetic + real world data sets.

**Strengths:**

-Motivating the proposed definition by surveying previous definitions is nice.
-The questions they seek to answer (detecting the violation of the proposed definition and how to enforce via post-processing) is nice.

**Weaknesses:**

-The actual proofs in the appendix seem like they can use a lot more explanations. For instance, the proof of Theorem 3 simply appeals to Shapiro’s result without almost any reference to the original optimization problem it’s considering: I.e. what exactly is the constraint part of the optimization, why does the convergence assumption in Theorem 3 (||\hat{pi}(x) - \pi_a(x)|| …. \le o(n^{-1/2})) imply the convergence in the estimation of the constraints in the optimization problem, etc? It would be helpful for the readers if these things are explained.


-I have some confusion about the doubly robust convergence assumption that’s made in Theorem 2 and 3. See the question below, but if it’s the case that the convergence is asking for convergence for each x not averaged over x, then the convergence assumption seems way too strong. Actually, staring at the optimization problems, I think the last two constraints that need to be estimated seem to be for each x not averaged over x, meaning one truly needs a convergence for every x? Is this because the fairness definition is binding at the individual level at every x?

**Questions:**

-Main theorems seem to say you need some form of uniform convergence for each x (Theorem 2, 3). But briefly skimming at the proof, it’s actually not asking convergence for each x, but rather averaged over the distribution D? More specifically, explaining what the notation ||\hat(pi)(x) - pi_a(x)||_2 means will clarify this confusion. I’m assuming it means \sqrt{E_{x \sim D}[(\hat(pi)(x) - pi_a(x))^2]}? If that’s not the case and it’s asking for uniform convergence for each x, then asking for such uniform convergence rate of 1/n^{1/2} for every x seems way too strong.

---

### Meta-Review · Area_Chair_mpoW · 2023-12-10

**Metareview:**

The paper proposes an alternative definition to counterfactual fairness, principled counterfactual fairness, which is applicable to those individual for which the sensitive information does not have an effect in their outcome. While the authors motivate well their notion, the reviewers raised important concerns that were not addressed during the rebuttal period. Thus, I encourage the authors to improve their paper based on the reviewers' comments.

**Justification For Why Not Higher Score:**

Important issues raised by the reviewers have not been addressed by the authors.

**Justification For Why Not Lower Score:**

N/A

---

### Decision · Program_Chairs · 2024-01-16

Reject